# Erythromycin Scavenging from Aqueous Solutions by Zeolitic Materials Derived from Fly Ash

**DOI:** 10.3390/molecules28020798

**Published:** 2023-01-13

**Authors:** Agnieszka Grela, Joanna Kuc, Agnieszka Klimek, Jakub Matusik, Justyna Pamuła, Wojciech Franus, Kamil Urbański, Tomasz Bajda

**Affiliations:** 1Faculty of Geology, Geophysics and Environmental Protection, AGH University of Science and Technology, 30-059 Cracow, Poland; 2Faculty of Environmental and Power Engineering, Department of Geoengineering and Water Management, Cracow University of Technology, 31-155 Cracow, Poland; 3Faculty of Chemical Engineering and Technology, Cracow University of Technology, 31-155 Cracow, Poland; 4Faculty of Civil Engineering and Architecture, Department of Construction Materials Engineering and Geoengineering, Lublin University of Technology, 20-618 Lublin, Poland

**Keywords:** adsorption, erythromycin, zeolite from fly ash, carbon–zeolite composites, wastewater, removal, Behnajady–Modirshahl–Ghanber (BMG) kinetic model

## Abstract

Erythromycin (EA) is an antibiotic whose concentration in water and wastewater has been reported to be above the standard levels. Since the methods used so far to remove EA from aquatic environments have not been effective, the development of effective methods for EA removal is necessary. In the present study, fly ash (FA)-based zeolite materials, which have not been investigated as EA sorbents before, were used. The possibilities of managing waste FA and using its transformation products for EA sorption were presented. The efficiency of EA removal from experimental solutions and real wastewater was evaluated. In addition, the sorbents’ mineral composition, chemical composition, and physicochemical properties and the effects of adsorbent mass, contact time, initial EA concentration, and pH on EA removal were analyzed. The EA was removed within the first 2 min of the reaction with an efficiency of 99% from experimental solutions and 94% from real wastewater. The maximum adsorption capacities were 314.7 mg g^−1^ for the fly ash-based synthetic zeolite (NaP1_FA) and 363.0 mg g^−1^ for the carbon–zeolite composite (NaP1_C). A fivefold regeneration of the NaP1_FA and NaP1_C showed no significant loss of adsorption efficiency. These findings indicate that zeolitic materials effectively remove EA and can be further investigated for removing other pharmaceuticals from water and wastewater.

## 1. Introduction

Every year, the consumption of drugs increases; this is mostly due to the storing and utilization of unused drugs. The presence of pharmaceuticals has been detected in surface water, groundwater, marine water, soil, and drinking water [1,2]. One group of pharmaceuticals that poses a major threat, primarily to aquatic environments, is antibiotics. The global consumption of antibiotics is 100–200 thousand tons per year [3]. Antibiotics and their metabolites are water pollutants which create a growing and worsening problem due to their low degradation rate. Their decomposition time can vary from months to even several years depending on several parameters [4,5]. Antibiotic concentrations in water and wastewater have been found to reach values of around µg L^−1^ [6,7,8]. Depending on the chemical properties of the antibiotic, about 5–90% of the absorbed antibiotic dose is excreted as a metabolite or a parent compound [9] and then ends up in water [10]. Techniques used in municipal wastewater treatment plants aim to reduce the concentrations of suspended solids, organic carbon, heavy metals, and nutrients such as nitrogen and phosphorus. This is achieved by using a combination of mechanical, chemical, and biological processes. However, these processes are not capable of eliminating antibiotics from aqueous solutions. Antibiotics in wastewater can adversely affect biological treatment processes, in which microorganisms are essential for the desired functioning of the treatment plant. Many previous studies [11,12,13] have proven that the efficiency of the removal of antibiotics is not sufficient in wastewater treatment facilities. Therefore, these facilities are considered the primary sources of the environmental pollution caused by antibiotics. The effluents produced contain several metabolites and transformation products of antibiotics that are formed in addition to the parent compounds via various chemical reactions [14].

Inadequately treated municipal wastewater is one of the major causes of surface water pollution [15]. The presence of antibiotics in surface water can lead to adverse effects in aquatic organisms [16,17]. Recent studies have shown that several antibiotics are toxic to some organisms and thus pose a threat to the environment, ecological balance, and aquatic life [18]. This is because antibiotics affect, among others, the endocrine and genetic systems of aquatic organisms [19,20]. Moreover, the presence of antibiotics in drinking water has also been confirmed [21].

The prevalence of contaminants such as antibiotics in surface water and groundwater has also been recognized by the Environmental Protection Agency [22]. Erythromycin (EA) is an antibiotic with concentrations above standard levels in water and wastewater [23,24,25,26,27,28]. Therefore, the efficient removal of EA from wastewater is highly important. This subsequently requires new approaches and technologies that will prevent the accumulation of EA in the natural aquatic environment and reduce its adverse effects on human health and the ecosystem.

EA is classified as a first-generation nonpolyene macrolide antibiotic. Discovered and isolated by J. M. McGuire in 1952, it was introduced to the market in 1965. A strain of the Gram-positive bacterium *Saccharopolyspora erythraea* is responsible for obtaining EA; however, biosynthesis produces a mixture of EA isomers, i.e., EA A, B, and C. The name “EA” is reserved for the main product of biosynthesis, i.e., the A isomer (which has the highest antimicrobial activity), with the proportion of the B and C isomers making up no more than 5% each [29,30].

The basic element of the EA molecule is a 14-membered, macrocyclic lactone ring that does not contain double bonds. This is the aglycone part, called erythronolide. The molecule also contains two sugar substituents: 6-deoxyhexose derivatives, which are attached via glycosidic bonds to the ring. EA shows a slightly basic character (pK_a_ = 8.89) and has five hydroxyl groups in its structure, which affects its affinity toward water (Table 1) [30].

The properties of EA highly depend on pH. In slightly alkaline solutions, its antibiotic activity is several times higher, whereas it is unstable in an acidic environment. In the latter case, it undergoes intramolecular cyclization through the formation of bonds between the carbonyl group at position C9 and the hydroxyl groups at positions C6 and C12. Moreover, in an acidic environment, EA is rapidly degraded by intramolecular cleavage of the water molecule [31].

To address the problem of antibiotics in wastewater, many conventional and advanced technologies have been developed [32,33]. Oxidation, electroprecipitation, membrane separation, coagulation–flocculation, evaporation, floatation, and ion exchange have been widely used, but these methods cannot effectively remove antibiotics from wastewater [23,24,34].

Adsorption has been identified as the best method for the removal of antibiotics [35]. It has the following advantages: less waste generation, suitability for both intermittent and continuous purification methods [36], applicability at low contaminant concentration levels, the possibility of regeneration and reuse, ease of operation, and low overall cost [37]. In addition, new low-cost adsorbents [38] such as activated carbon or nanoparticles that are used in treatment systems can be applied in the adsorption process [39]. For example, granular activated carbon (GAC) and powdered activated carbon (PAC) are the most commonly used materials to remove antibiotics via adsorption in secondary treatment systems in order to avoid the formation of transformation by-products [40,41]. Recent studies have shown that walnut shell-based activated carbon shows a high adsorption potential for metronidazole and sulfamethoxazole antibiotics, with maximum adsorption capacities of 107.4 and 93.5 mg g^−1^, respectively [42]. Micrograin GAC was used in a large-scale pilot experiment in a wastewater treatment plant in France. Thanks to the fluidized bed of activated carbon, both conventional parameters of the treated effluent were improved, and the concentrations of micropollutants were significantly reduced. Of the 31 pharmaceuticals and hormones analyzed, 26 were reduced by more than 50%, and compounds considered to be performance indicators of the treatment process were degraded by 78–89% [43]. PAC was also used to simulate water purification processes from endocrine-disrupting compounds such as pharmaceuticals and personal care products. After the use of PAC, almost one third of the 62 compounds were degraded to a high degree (>90%) and only eight had less than 50% removed [44].

Zeolites synthesized from fly ash (FA) and zeolite–carbon composites meet the aforementioned criteria and can be used in adsorption processes. Zeolites have a three-dimensional crystal structure comprising AlO_4_ and SiO_4_ tetrahedra, which form cages and channels [45,46]. The crystal structure of zeolites is stable, allows for ion exchange, and accommodates water molecules, cations, and organic molecules [47,48]. These materials are characterized by high porosity and a large surface area. The shape of the pores directly affects their adsorption selectivity toward host molecules [49].

This study aimed to use zeolite (NaP1_FA) and zeolite–carbon composite (NaP1_C) as potential adsorbents for EA removal. The studied adsorbents were produced from FA formed during coal combustion, which is an attractive alternative to ash. This makes the adsorbents low-cost materials that can be used to remove antibiotics from aqueous solutions [50,51,52]. The effects of many parameters, including contact time, adsorbent dose, and concentration, were investigated. The scanning electron microscope (SEM), N_2_ adsorption/desorption, x-ray diffraction (XRD), Fourier transform infrared (FTIR) spectroscopy, and thermogravimetry differential thermal analysis (TG/DTA) were used to characterize the studied adsorbents.

The novelty of this work involves the use of two different fly ash-based adsorbents that show a high EA removal efficiency from both artificial and real wastewater. Moreover, these materials can be easily regenerated and reused without a significant decrease in their adsorption efficiency. In addition, this experimental study will undoubtedly increase the knowledge required for conducting further studies on a larger scale. The results of EA removal presented in this study were achieved for the first time using the sorbents NaP1_FA and NaP1_C.

## 2. Results and Discussion

### 2.1. Characterization of Adsorbents

#### 2.1.1. XRD and FTIR Analysis

The synthesis of zeolites from the FA of different carbon contents resulted in the formation of zeolite NaP1_FA and zeolite–carbon composite NaP1_C (Figure 1a). The diffractograms showed the presence of NaP1 zeolite as the main phase, with d spacing values of 7.10, 5.01, 4.11, and 3.18 Å, which was accompanied by admixtures of unreacted FA components, including mullite and quartz. The presence of amorphous substances, probably including amorphous silica in the NaP1_FA sample and carbon-rich material in the NaP1_C sample, was recognized by characteristic background increases ranging from 10 to 35°2θ. Previous studies have reported a similar mineral composition of zeolite NaP1_FA and zeolite–carbon composite NaP1_C [53,54,55].

The FTIR spectra confirmed the observations made by XRD (Figure 1b). A wide, broad band of 3600 to 3100 cm^−1^ for both materials corresponded to the stretching vibrations of hydroxyl groups from the adsorbed water molecules in zeolite channels [56,57]. All materials showed clear bands at approximately 1644, 1424, and 738 cm^−1^. These can be attributed to the bending vibrations of H–OH (indicating that the samples contain a fraction of free-state water adsorbed on the surface or inside the structural cavities), the O–C–O stretching vibration associated with carbonates and the stretching vibrations of the Si–O–Si/Si–O–Al bonds in the tetrahedra. A band at around 1090 cm^−1^ registered for NaP1_FA and NaP1_C may be attributed to the stretching vibration of C–OH. The bands at 565–567 cm^−1^ corresponded to the mullite admixture. As reported by [58], the presence of mullite is also confirmed by the presence of bands at 1099 and 811 cm^−1^. In the present study, the bands responsible for mullite were observed at 1085 and 802 cm^−1^ (NaP1_FA), and 1093 and 798 cm^−1^ (NaP1_C). The bands at 734–738 cm^−1^ and 604 cm^−1^ were attributable to Si–O–Si/Si–O–Al and Al–O/Si–O bending vibrations, respectively [58]. As reported by [59], the bands at 738 and 690 cm^−1^ (weak band visible in Figure 1b) in the P1 zeolite are attributable to symmetric stretching vibrations of the internal tetrahedron. These authors have also stated that the band centered at 605 cm^−1^ (Figure 1b—NaP1_FA) corresponds to the presence of a double ring in the P1 zeolite. The band vibration of the TO_4_ (T = Si or Al) tetrahedral in zeolite P1 was observed at 437 cm^−1^ (Figure 1b). These peaks agree well with the literature and confirm the successful synthesis of zeolite P1 [60,61]. Most importantly, a sharp band of high intensity at around 990 cm^−1^ observed for NaP1_FA and NaP1_C confirmed the formation of a well-defined aluminosilicate skeleton. These bands can be attributable to the asymmetric stretching vibrations of bridge bonds Si–O(Si) and Si–O(Al) in the zeolite frameworks.

#### 2.1.2. DTA/TG and Carbon, Hydrogen, and Nitrogen Elemental Analysis

The thermal curves of NaP1_C and NaP1_FA enabled us to evaluate the differences in their thermal behavior. The first weight loss (up to 6 wt.%) of NaP1_C and NaP1_FA started in the first stage of heating to around 200 °C and was attributable to the loss of water adsorbed on the surface of the materials [55]. In the case of the materials obtained, the loss of adsorbed water was comparable and reached around 6 wt.% for NaP1_FA and 4 wt. % for NaP1_C. Similar water loss results (about 4% for NaP1_C) on the TG curve were reported by Bandura et al. [55], who assessed the ability of NaP1_C and NaP1_X FAs to adsorb petroleum substances.

Woszuk et al. [62] conducted a thermal analysis in which the effect of NaP1_FA and clinoptilolite as additives to asphalt materials was investigated. Endothermic effects (about 100 °C) and mass loss reaching a maximum of 6% for the NaP1_FA sample were compared with the data in Figure 2(1a,2a).

The DTA curves showed that the kinetics of water loss differed between the materials. For NaP1_C, the water release was slower but extended (from 25 to 200 °C), whereas for NaP1_FA, it was rapid and was finished at around 180 °C. The QMS indicated the release of water. The next decomposition effect started at 420 °C and led to a weight loss of ~36 wt.% for NaP1_C and ~4 wt.% for NaP1_FA. It can be due to carbon decomposition present in the materials, which is accompanied by a CO_2_ QMS signal [63]. The carbon content is in close agreement with the CHN analysis: NaP1_FA (C—5.53 wt.%) and NaP1_C (C—42.19 wt.%). The carbon combustion ends at 790 °C for NaP1_C and 825 °C for NaP1_FA. Woszuk et al. [62] and Zhou et al. [64] have reported an exothermic effect in the range of 450–800 °C for the thermal curves of NaP1_FA zeolite. In the present study, this range started at 300 °C and ended around 800 °C, which is attributable to the dehydroxylation of the OH group. The DTA curves showed exothermic peaks related to the oxidation of carbon species with maxima for NaP1_FA at 570 and 675 °C and for NaP1_C at 435 °C. The total weight loss is usually related to the carbon content; however, some minor contribution of dehydroxylation is possible. Above 900 °C, the synthesis of new phases took place, which was accompanied by an exothermic effect. The total mass losses for NaP1_C and NaP1_FA were ~42 and ~14 wt.%, respectively (Figure 2). Zhou et al. [64] obtained similar results for the total weight loss for the NaP1_FA sample, which was about 10% and, additionally, as a result of the decomposition of carbonates.

#### 2.1.3. Texture and Morphology

For the analyzed materials, type IV isotherms were identified according to the IUPAC classification (Figure 3a,b). A hysteresis loop belonging to the H3 type reflected the presence of slit-shaped mesopores. This shape of isotherms was consistent with those often found for aggregated crystals of zeolites and micromesoporous carbons [65]. The SBETs of NaP1_FA and NaP1_C were similar and approximately 60 m^2^ g^−1^. However, the NaP1_FA zeolite P1 was characterized by a larger volume of mesopores.

The BJH pore size distribution of NaP1_FA showed a domination of 2–3-nm pores, with an average pore diameter of approximately 8.7 nm. The P1-C pore size distribution showed a dominance of pores on the border between micropore and mesopore sizes. The average pore diameter was approximately 7.8 nm (Table 2).

The morphology observations of crystals allowed the identification of the characteristic representatives of zeolites (Figure 3e,f). The SEM observations of the agglomerates showed that the NaP1_FA zeolite material formed sphere-like aggregates built from plate-like particles with a length of ~0.5–1.0 µm. Such morphology is characteristic of the gismondite group of zeolites [66]. During the synthesis of NaP1_C and the consolidation process, the addition of cement (high-carbon ash) and water slightly changed the structure, making it more compact. Furthermore, zeolite crystals were noticeable in the structure.

### 2.2. EA Adsorption Experiments

#### 2.2.1. Evaluation of Maximum Adsorption Capacity

The adsorption isotherms were measured to evaluate the adsorption capacities of the NaP1_FA and NaP1_C samples (Figure 4). The obtained experimental data were fitted using the Langmuir and Freundlich isotherm models (Table 3). The calculated adsorption parameters are summarized in Table 4.

The adsorption capacity reached 5.66 and 7.14 mg g^−1^ for the NaP1_C sample according to the Langmuir and Freundlich models, respectively. However, for the NaP1_FA sample, it reached 1.19 and 5.72 mg g^−1^ for the two models, respectively. Much higher concentration values were obtained using the Freundlich model for the NaP1_C sample. In all cases, the regression coefficient (R^2^) of the linear fit for both samples was >0.74.

The R^2^ values showed that the Langmuir model fits the results slightly better than the Freundlich model, which is in agreement with earlier studies regarding EA adsorption by zeolites [46].

#### 2.2.2. The Effect of pH and Dosage

The experiments performed at different initial pH values allowed us to evaluate the changes in EA removal efficiency by the two zeolitic adsorbents (Figure 5a). A decline in the efficiency for the NaP1_FA material was observed following an increase in pH. Under acidic conditions (initial pH of 2.0), the removal was around 80%, whereas it was decreased to around 30% for a pH of 8.0. In contrast, pH did not affect EA removal by NaP1_C. A removal of around 80% was observed regardless of pH conditions. This may indicate that hydrophobic interactions dominate in the reaction of EA with NaP1_C, which is rich in carbon. On the other hand, larger amounts of the hydrophilic surface of NaP1_FA being available results in polar interaction with EA, which is sensitive to pH.

Based on the analysis of EA equilibrium concentrations (µg L^−1^), it can be concluded that the highest adsorbent doses of 2.0 g L^−1^ resulted in the removal of EA to the greatest extent (Figure 4b). EA concentration after adsorption on NaP1_FA and NaP1_C decreased to 35.5 and 23.5 µg L^−1^, respectively. In previous studies, the adsorption capacity expressed in mg g^−1^ was used as a measure of adsorption efficiency. The adsorbent dose of 2.0 g L^−1^ used in this study resulted in EA capacities of 11.75 mg g^−1^ (NaP1_FA) and 17.75 mg g^−1^ (NaP1_C). For example, for the magnetic activated carbon (MACC) adsorbent (nanocomposites prepared using the coprecipitation method), adsorption values of 178.52 mg g^−1^ were achieved using a dose of 0.001 g L^−1^, resulting in 54% EA removal [67]. In the present study, 61% EA removal was achieved by NaP1_C and 74% by NaP1_FA. Experiments performed on EA removal with the adsorbent ZCA (zeolite/cellulose acetate blend fiber) showed that the adsorption capacity of ZCA is almost equal to that obtained in the present study for NaP1_C, which was 17.76 mg g^−1^ with an efficiency of 97% [68]. EA has also been removed using BSA/Fe_3_O_4_ (magnetic composite microspheres) and MWCNT (multiwalled carbon nanotubes), whose adsorption capacities were 144.51 and 104.8 mg g^−1^, respectively. The efficiencies of the EA removal processes from aqueous solutions were higher than in the present study: 93.71% for BSA/Fe_3_O_4_ and 83.71% for MWCNT [69,70]. Another adsorbent is MACC (nanocomposites obtained using the coprecipitation method), whose adsorption values of 178.52 mg g^−1^ were achieved using a dose of 0.001 g L^−1^, resulting in 54% EA removal [67]. The literature contains information on the use of porous magnetic graphene (PMG) and EA molecularly imprinted polymer (ERY@MIP) in the purification of EA aqueous solutions, which have adsorption capacities of 286 and 44.03 mg g^−1^, respectively [71,72]. In previous studies on EA removal, magnetic activated carbon adsorbents were used, with an adsorption capacity of 248.91 mg g^−1^ for a dose of 1.55 g L^−1^ [73]. EA was also removed using the nanomaterials MoS_2_ (molybdenum disulfide) and WS_2_ (tungsten disulfide). For doses of 1.0 g L^−1^, adsorption capacities of 57.10 mg g^−1^ for MoS_2_ and 190.31 mg g^−1^ for WS_2_ were obtained [74].

In recent years, EA removal has been studied using various materials (Table 5). In Table 5, a summary of NaP1_FA and NaP1_C adsorbents with significant adsorption efficiencies against EA is presented. This comparison shows that FA zeolite and zeolite–carbon composite can be alternatives to the other materials studied so far.

The dependency of the surface charge density of NaP1_FA and NaP1_C on solution pH value is presented in Figure 6. Potentiometric titrations indicated that the pH ZPC of both sorbents is about 3.0. This indicates that at a pH of 3.0, their surfaces are characterized by zero surface charge. However, while at pH < 3.0 the solid surface is positively charged, at pH > 3.0, it is negatively charged. At a pH higher than 3.0, both sorbents are negatively charged, and the surface negative charge density of NaP1_FA is almost twice that of NaP1_C at the same pH value. This relationship is correlated with the EA sorption efficiency, which is half for NaP1_FA of that of NaP1_C in alkaline media. EA has a pKa of 8.89, indicating that most of its molecules would be positively charged at pH < 8.89. This explains the relatively high sorption on negatively charged zeolite materials under these pH conditions. However, the electrostatic attraction seems to play a secondary role in the removal process, as NaP1_C shows a higher efficiency. This indicates the dominant role of hydrophobic interactions between the carbon component present in NaP1_C and the organic structure of EA. The latter is rich in methyl and methylene groups, thus leading to its hydrophobic nature. Such a mechanism is independent of solution pH. The more hydrophilic nature of NaP1_FA compared to NaP1_C results in lower P1 zeolite sorbing of EA, which is significantly worse than the zeolite–carbon composite in alkaline media adsorption. On the one hand, the negative surface charge of NaP1_FA is twice that of NaP1_C, resulting in lower sorption. On the other hand, the carbon in the composite is responsible for the hydrophobic interactions, resulting in a less pH-dependent EA sorption on NaP1_C than on NaP1_FA.

A model showing the possible interactions of EA with zeolitic materials is presented in Figure 7. It shows two types of mechanisms: (i) the electrostatic interaction of protonated EA with the negatively charged surface of zeolites and (ii) the hydrophobic interaction of EA with the carbon component present in NaP1_C. In particular, the latter mechanism is responsible for highly efficient adsorption.

#### 2.2.3. Kinetics

To best reproduce the kinetics of EA adsorption on selected adsorbents, four kinetic models were analyzed: the pseudo-zero-order, pseudo-first-order, pseudo-second-order, and Behnajady–Modirshahl–Ghanber (BMG) kinetic models.

The BMG kinetic model can be described as follows:(1)$$\frac{c_{t}}{c_{0}}=1-\frac{t}{m+b}$$
where *c_t_* is the concentration of EA at time *t* (µg L^−1^), *t* is the duration of the experiment (min), *c*_0_ is the initial concentration of EA (µg L^−1^), and *m* and *b* are characteristic constants related to reaction kinetics and EA degradation capacity, respectively.

Subsequently, by plotting $\frac{t}{\frac{1-c_{t}}{c_{0}}}$ depending on *t*, a straight line is obtained with the slope b and the intersection with the ordinate axis at the point *m* [75].

The other three kinetic models—the pseudo-zero-order, pseudo-first-order, and pseudo-second-order models—were not successful. This was confirmed by a statistical evaluation (Table 6), i.e., the low value of the correlation coefficient (R^2^). Only the BMG kinetic model allowed for the interpretation of EA kinetics and produced high R^2^ values.

In line with the above assumptions, curves representing the BMG kinetic model were drawn (Figure 8). These curves were supplemented with values of EA retention by the adsorbents, which were expressed in percentages.

A high removal rate is essential for practical applications of solid-state adsorbents in dynamic systems. For both adsorbents, a fast uptake of EA was observed; the equilibrium was reached in the first 2 min of the reaction (Figure 8). Regardless of the adsorbent type, the correlation coefficient (R^2^) was one, which indicates a high agreement of the BMG kinetic model to the experimental data. The reciprocal of the constant b was the theoretical maximum amount of EA that could be removed [75]. The 1/b values were 0.9924 and 0.9915 mg g^−1^ for NaP1_C and NaP1_FA, respectively. This indicates that the adsorption, expressed in a percentage, should be over 99%. The estimated value of the amount of adsorbed EA was consistent with that obtained in the experiments. In line with the assumptions of the BMG model, the higher the value of 1/m, the faster the pollutant removal in the starting minutes [75]. Based on the inverse of the m parameter, the initial adsorption for NaP1_C was four times faster than that for NaP1_FA.

The BMG kinetic model is not widely used in the description of the kinetics of pollutant removal studies by adsorption. According to reports in the literature, for adsorption experiments with the removal of pharmaceuticals, the kinetics of the degradation of compounds is usually described using a pseudo-second-order model [76]. The BMG model was used primarily to describe the reaction kinetics for the removal of hardly degradable pharmaceuticals from wastewater using the Fe(0)–catalytic Fenton oxidation methods [77] electro-oxidation and solar electro-oxidation [78]. Dyes form another type of pollutant that is difficult to remove from water. Mushtaq et al. proposed the use of carbon FA nanocomposites for the preparation of a photocatalyst that enables the degradation of micropollutants. The kinetics of discoloration is also best represented using the BMG model [79].

#### 2.2.4. Regeneration Studies

The regeneration possibility and stability of NaP1_FA and NaP1_C adsorbents are important for their future use. To evaluate the reusability of these adsorbents, a total of five adsorption/desorption cycles were conducted, and the results are presented in Figure 9. The adsorbents were rinsed once with ethanol after each adsorption cycle and then reused for the next adsorption step [73,80]. The results showed that there is no significant decrease in EA removal efficiency after five cycles of regeneration, suggesting that both adsorbents are stable and can be further reused.

#### 2.2.5. Efficiency in the Treatment of Real Wastewater

A dose of 0.5 g L^−1^ and a contact time of 60 min were used in the experiments on EA removal from wastewater (wastewater chemistry and parameters presented in Table 7). The EA concentration in the effluent before adsorption onto NaP1_FA/NaP1_C was 96.78 µg L^−1^. After adsorption on NaP1_FA and NaP1_C, it decreased to 15.90 and 5.38 µg L^−1^, respectively. Thus, NaP1_C showed a higher efficiency of more than 90% removal of EA from the treated wastewater than NaP1_FA.

Previous studies on EA removal have also been conducted using synthetic zeolites (zeolite MOR400 with a SiO_2_/Al_2_O_3_ ratio of 400 and zeolite Y) and natural zeolite from the Slovakian company Zeocem a.s., with different fractions (200 mm, 0.5–1 mm, and 1–2.5 mm). EA adsorption occurred throughout the zeolite structure as well as in the micropores [46]. The removal of EA from wastewater (concentrations: 16.0 ng L^−1^ Stupava Wastewater Treatment Plant (WWTP); 37.0 ng L^−1^ Devínska Nová Ves WWTP) using the Zeocem a.s. natural zeolite with three fractions was carried out using a 30-min contact time, and more than 90% removal was achieved for both WWTPs. According to studies conducted on wastewater, such a high efficiency was attributable to optimal pH conditions, which must be lower than the EA’s pKa, i.e., 8.88. The highest efficiency of EA removal from wastewater was achieved using zeolites with the smallest fraction of 200 µm (WWTP Stupava: EA 94.7%, WWTP Devínska Nová Ves: EA 98.5%). These results confirm that zeolites can be used as sorbents to remove EA from wastewater [81]. For zeolite Y, studies analyzing the EA adsorption efficiency were conducted using water collected at the outlet of a wastewater treatment plant in Ferrara (northern Italy), in which the actual EA concentration was 1.10 µg L^−1^. The results confirmed that 100% removal was achieved using this zeolite [46].

## 3. Experimental Method

### 3.1. Materials and Chemicals

Zeolite NaP1_FA was synthesized from FA following a previously described method [53,54,82]. Zeolite NaP1 was prepared based on the hydrothermal synthesis (for 24 h at 353 K) of FA with sodium hydroxide at atmospheric pressure. The zeolite–carbon NaP1_C composite was synthesized according to a previously reported procedure [55]. Briefly, 200 g of carbon-rich FA (HiC) was mixed with 1000 mL of 3 M NaOH solution. The mixture was stirred at 80 °C for 48 h. Then, the material was rinsed six times with 500 mL distilled water and dried at 105 °C. The major difference between NaP1_FA and NaP1_C is the use of FA with a different unburned carbon content for their synthesis. To synthesize NaP1_FA, F-class FA from coal combustion at the Rybnik Power Plant in Poland, which contains < 5% of unburned carbon, was used [82]. For the synthesis of NaP1_C, high-carbon FA collected from the Janikowo Thermal Power Station (Janikowo, Poland) was used [55], which contains about 40% unburned carbon. The carbon contents of the two materials after synthesis were different: 5.53 and 42.19% for NaP1_FA and NaP1_C, respectively.

EA was purchased from Sigma-Aldrich. Stock solutions at a concentration of 1.0 µg mL^−1^ were prepared by dissolving EA in acetonitrile. The prepared solutions were stored in a refrigerator (for no longer than 6 months). Acetonitrile LC-MS Chromasolv^®^ (≥99.9%) was purchased from Sigma-Aldrich, and ammonium acetate p.a. (≥99%) was obtained from Fluka. All reagents were of analytical grade. The solutions for analysis were filtered using a 0.22-µm syringe filter with a nylon membrane.

The wastewater used in the adsorption experiments was collected from the Polish Municipal Wastewater Treatment Plant (Table 7).

### 3.2. Adsorption and Desorption Experiments

For the batch experiments, two samples were chosen: NaP1_FA and NAP1_C adsorbents. Unless stated otherwise, all experiments were run in duplicate, the solid/solution ratio was set to 1.0 g L^−1^, and after 1 h of reaction time, the suspensions were centrifuged (4500 rpm for 10 min) and filtered through a 0.22-µm syringe filter before undergoing high performance liquid chromatography (HPLC) analysis. In addition, prior to reactions with EA, all solid adsorbents were conditioned by shaking them in water for 1 h. In all cases, the equilibrium pH was measured.

First, EA adsorption experiments were performed on the 100 µg L^−1^ EA concentration to analyze the differences in removal efficiency between the adsorbents. Second, the full adsorption isotherms were determined at a concentration of 0.1–200 mg L^−1^. The samples from the above experiments were subjected to desorption using 20 mL ethanol. For the samples that reacted with a 100 µg L^−1^ EA concentration, five subsequent cycles of regeneration were performed with ethanol. Third, the pH effect was analyzed for the initial EA concentration of 100 µg L^−1^ and initial pH values of 2.0, 2.5, and 8.0. Using the potentiometric titration method, both the surface charge density as a function of solution pH and point of zero charge were determined. The analyzed suspensions were prepared using 100 mg of the solid and 50 mL of the electrolyte (0.001 M NaCl). The systems were titrated with 0.1 M NaOH in the pH range of 2.9–10. Titrations were performed using an Eco Titrator (Metrohm). The solid surface charge density (σ0) [C m^−2^] was calculated based on the following equation [63]:σ0 = (ΔV · c · F)/(m · S),
where ΔV is the difference in the base volume added to a suspension and a supporting electrolyte solution that leads to the specific pH value [ml], c is the base concentration [mol L^−1^], F is the Faraday constant, m is the activated sorbent in the suspension [g], and S is the solid surface area [m^2^ g]. Fourth, the dosage effect experiment was carried out with adsorbent doses equal to 0.1, 0.2, 0.5, 1.0, and 2.0 g L^−1^. Fifth, the EA adsorption kinetics were evaluated for 100 µg L^−1^ initial concentration and contact times in the range of 0.5–360 min.

### 3.3. Analytical Methods for Solid-State Characterization

The XRD patterns of the powdered samples were recorded in the range of 2–75° 2θ (0.05° 2θ increments) using a Rigaku MiniFlex diffractometer with CuKα radiation (λ = 1.5418 Å). The FTIR spectroscopy analyses were carried out using a Thermo Scientific Nicolet 6700 spectrometer in the transmission mode. The FTIR spectra were collected from standard KBr pellets (1 wt% sample/KBr) in the 4000–400 cm^−1^ range (64 scans with 4 cm^−1^ resolution). The morphologies of the samples were observed using a Quanta 200 FEG scanning electron microscope. The samples were analyzed in the low vacuum mode without prior surface coating. The thermal (DTA/TG) analysis of the samples was carried out using a Netzsch STA (Simultaneous Thermal Analysis) 449F3 instrument coupled with a Quadrupole Mass Spectrometer Netzsch (QMS) 403. The measurements were taken in a temperature range of 30–1000 °C (heating rate 10 °C/min, air atmosphere). The samples’ surface area and porosity were examined based on low-temperature N_2_ adsorption/desorption measurements (77 K) using a Micromeritics Accelerated Surface Area and Porosimetry System (ASAP) 2020 instrument. Prior to the analysis, the samples were outgassed at 105 °C for 12 h. The specific surface area (SBET) was calculated based on the BET equation. The total pore volume (V_tot_) was calculated from the amount of adsorbed nitrogen at a relative pressure close to 1.0. The volume of micropores (V_micro_) was calculated using the Dubinin–Radushkevich method. The volume of mesopores (V_meso_) was determined using the Barrett–Joyner–Halenda (BJH) method. The pore size distributions were determined from the desorption branch of the isotherm using the BJH procedure.

### 3.4. Liquid Sample Extraction and UPLC (Ultra Performance Liquid Hromatography)–MS/MS Analysis

#### 3.4.1. UPLC–MS/MS Determination Method

EA was determined using a Modular (U) HPLC system—Nexera (LC-40) series, Shimadzu, with a triple quadrupole analyzer and a QTrap 3200 reaction chamber (AB Sciex, Framingham, MA, USA). Electrospray ionization (ESI) and multiple reaction monitoring were applied.

An EA standard solution with a concentration of 1 µg ml^−1^ was used to optimize the ionization parameters and detection. After selecting the precursor ion, the collision energy was optimized to select the most appropriate (the highest MS/MS response signal) fragmentation pathway to produce ions. The following ions were selected: precursor ions of 734.6 m/z and product ions of 158.4 and 576.5 m/z for the identification and quantification of EA, respectively. The collision energy values for the selected ions were optimized using the SCIEX Analyst^®^ mass spectrometer software.

The chromatographic separation was performed in an isocratic system using a Kinetex 2.6 µm XB C18 100A, 100 × 2.1 mm column with a SecurityGuard Ultra cartridge C18 precolumn (both from Phenomenex). The mobile phase was a mixture of 0.2% formic acid in water LC-MS and 0.2% formic acid in acetonitrile LC-MS.

#### 3.4.2. EA Extraction from Municipal Wastewater

For the analyte extraction from the municipal wastewater matrix, solid-phase extraction (SPE) Oasis^®^ HLB (Hydrophilic-Lipophilic Balance) columns filled with divinylbenzene-co-N-vinylpyrrolidone (Waters) were used. The bed is stable within the pH range of 0.0 to 14.0; therefore, it is a universal cartridge for the isolation of acidic, neutral, and alkaline compounds.

In addition, it is necessary to use suitable analytical methods to determine the extent of environmental contamination with antibiotics such as EA. In most cases, liquid chromatography coupled with tandem mass spectrometry in the positive ESI mode (LC-ESI-MS/MS) was used. This method is powerful in terms of selectivity and sensitivity, allowing for the quantification of pharmaceuticals in water and wastewater at ng L^−1^ levels [30,83,84].

However, low concentrations of pharmaceuticals in water and wastewater require an SPE. This allows for the necessary preconcentration of the target analytes and purification from the interferents present in the sample matrices [85]. The Waters Oasis^®^ HLB is the most commonly used adsorbent in EA analysis. It is distinguished by its versatility, and it can be used for alkaline, neutral, and acidic compounds [86,87,88]. Lipophilic vinylbenzene and hydrophilic units present in the structure of N-vinylpyrrolidone allow this apparatus to be applied to a wide range of pH values—from 0 to 14 (Waters.com). Therefore, the details presented in this work are a valuable addition to this area of analytical methodology. The research methodology is presented in Figure 10.

## 4. Conclusions

EA removal efficiency depends on several factors, both physical (related to the degree of wastewater treatment) and chemical (attributable to the composition of the aqueous matrix). However, the most important parameter in antibiotic adsorption methodology is the choice of appropriate adsorbent composition and process conditions. The authors have performed numerous laboratory experiments, which confirm that NaP1_FA and NaP1_C adsorbents are capable of efficiently removing EA from aqueous solutions. The maximum adsorption capacity predicted by the Langmuir model was 1.19 mg g^−1^ for NaP1_FA and 5.66 mg g^−1^ for NaP1_C. Adsorption was pH-dependent for NaP1_FA. In this case, for a low pH of 2.0, the adsorption efficiency reached 80%, and for a higher pH of 8.0, it decreased to about 30%. The effect of pH on efficiency was not observed for NaP1_C, where the EA removal efficiency was consistently around 80%. NaP1_C showed good EA removal efficiency over a wide pH range. This is highly beneficial from an implementation perspective in the treatment stage of wastewater, whose pH varies but is usually weakly alkaline. EA removal is also dependent on the adsorbent dose. The best results were achieved for the highest dose of 2.0 g L^−1^. The EA adsorption efficiency of NaP1_FA reached 11.75 mg g^−1^, whereas for NaP1_C, it was 17.75 mg g^−1^. Both adsorbents removed EA rapidly, and equilibrium was reached within the first 2 min of the reaction. Furthermore, a regeneration experiment of NaP1_FA and NaP1_C using ethanol was also carried out successfully. The adsorbents showed more than 90% EA removal efficiency after five regeneration cycles. The authors also tested the performance of these sorbents using wastewater from a wastewater treatment plant. The results of this study indicated that NaP1_FA and NaP1_C are effective adsorbents that can be used for the removal of antibiotic contamination (EA) in wastewater remediation.

## Figures and Tables

**Figure 1 molecules-28-00798-f001:**
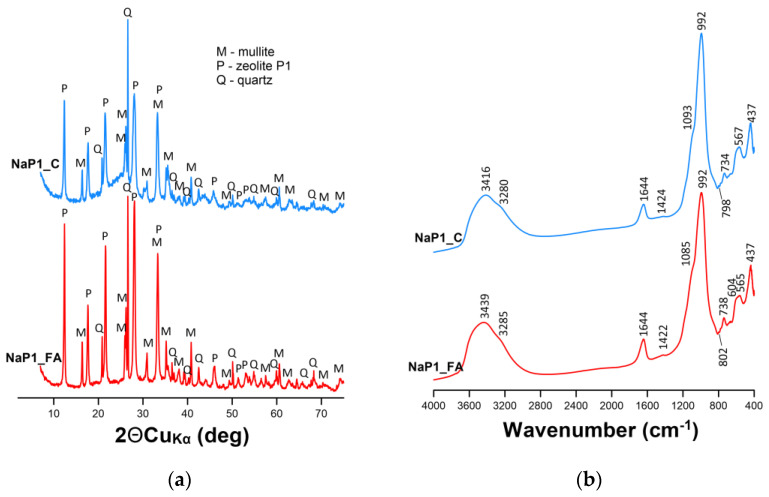
(**a**) X-ray diffraction patterns and (**b**) Fourier transform infrared spectra of NaP1_FA and NaP1_C samples.

**Figure 2 molecules-28-00798-f002:**
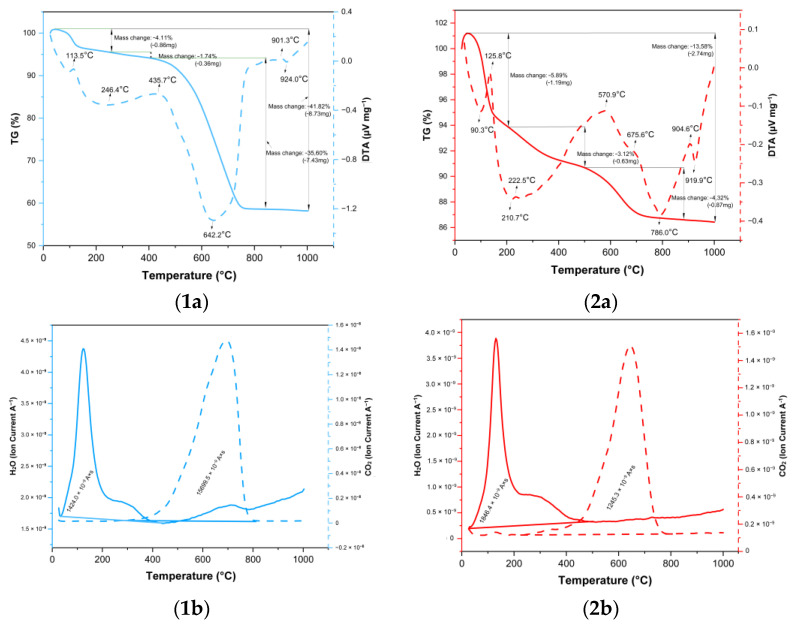
Thermal curves of (**1**) NaP1_C and (**2**) NaP1_FA [(**a**) TG/DTA, (**b**) QMS].

**Figure 3 molecules-28-00798-f003:**
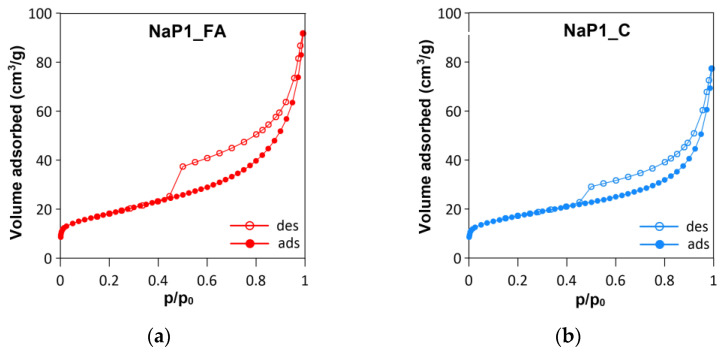
Textural analysis of NaP1_FA and NaP1_C samples: (**a**,**b**) N_2_ adsorption/desorption isotherms, (**c**,**d**) pore size distribution diagrams, and (**e**,**f**) SEM microphotographs.

**Figure 4 molecules-28-00798-f004:**
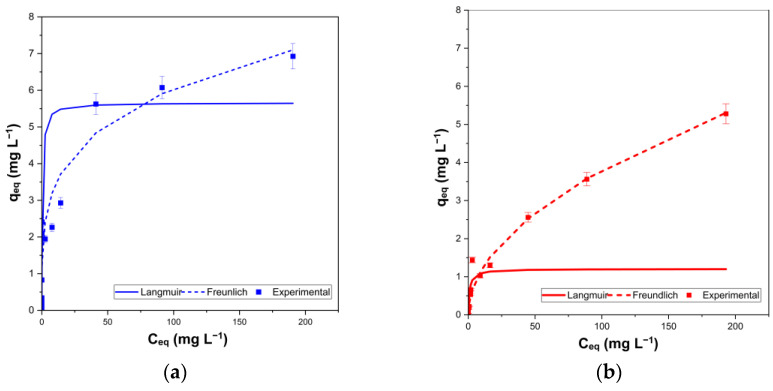
Adsorption isotherms for the investigated materials (**a**) NaP1_C and (**b**) NaP1_FA.

**Figure 5 molecules-28-00798-f005:**
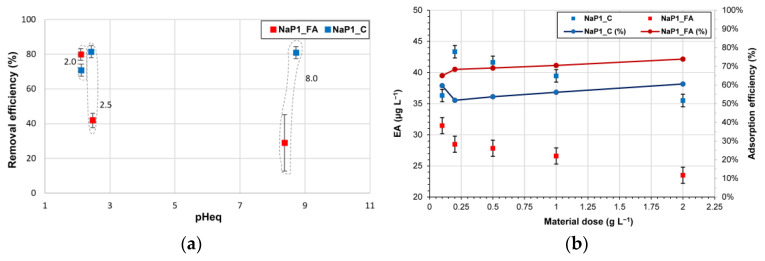
The effect of (**a**) pH and (**b**) dosage on erythromycin removal. The values on the graph indicate the initial pH.

**Figure 6 molecules-28-00798-f006:**
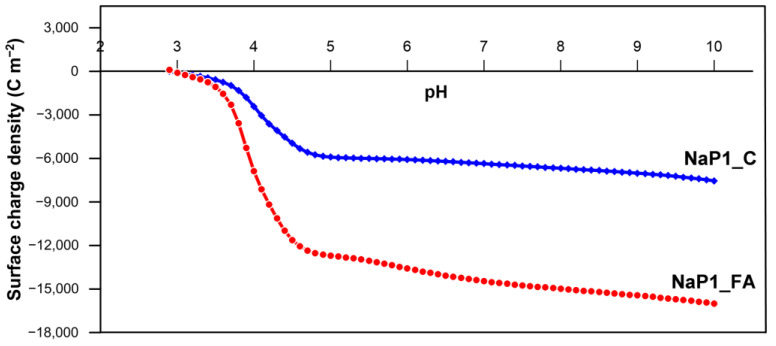
Surface charge density of NaP1_C and NaP1_FA as a function of solution pH.

**Figure 7 molecules-28-00798-f007:**
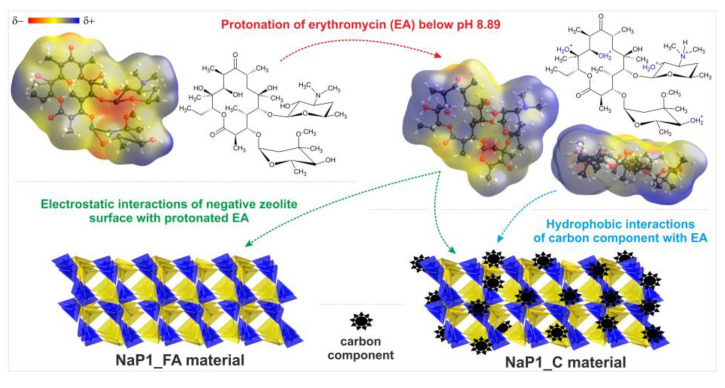
Possible interactions of erythromycin with zeolitic materials.

**Figure 8 molecules-28-00798-f008:**
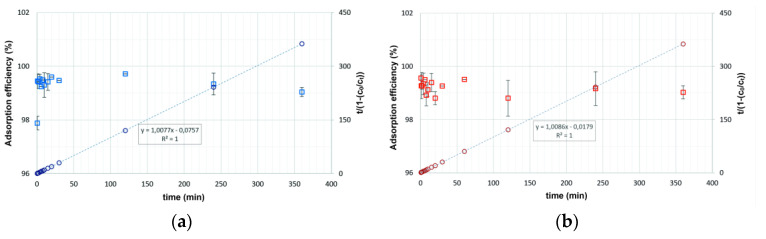
BMG kinetic models for the adsorption of EA on (**a**) NaP1_C and (**b**) NaP1_FA.

**Figure 9 molecules-28-00798-f009:**
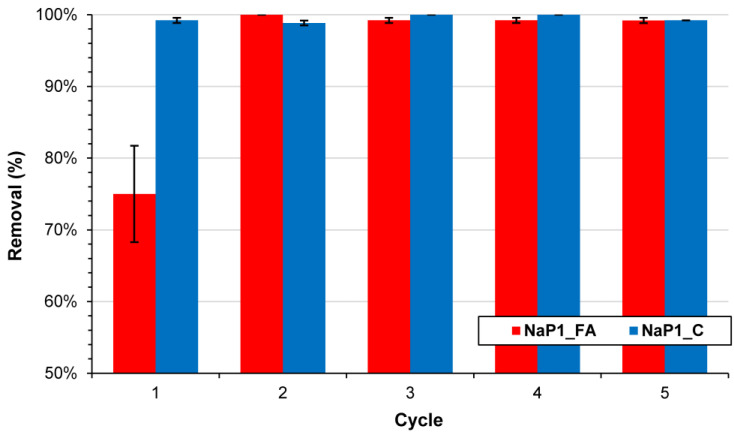
The EA removal efficiency of NaP1_FA and NaP1_C after regeneration cycles.

**Figure 10 molecules-28-00798-f010:**
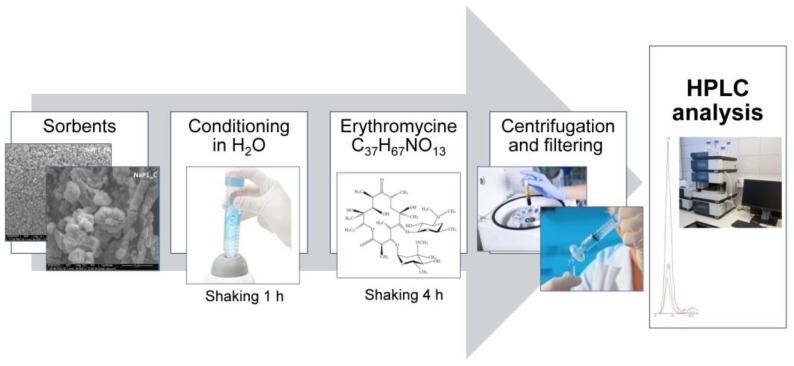
Research methodology flow chart.

**Table 1 molecules-28-00798-t001:** Physicochemical properties and chemical structure of erythromycin.

Therapeutic Class	Chemical Structure	pKa	Reference
Antibacterial agent	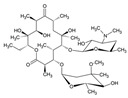	8.89	[29]

**Table 2 molecules-28-00798-t002:** The textural parameters of obtained zeolitic materials.

Material	S_BET_ m^2^ g^−1^	V_tot_^0.99^ cm^3^ g^−1^	V_mic_^DR^ cm^3^ g^−1^	V_mes_^BJH^ cm^3^ g^−1^
NaP1_FA	64.7	0.139	0.024	0.101
NaP1_C	61.0	0.115	0.023	0.078

**Table 3 molecules-28-00798-t003:** Langmuir and Freundlich models of equilibrium adsorption.

Model	Equation	Symbol Explanation
Langmuir	$\frac{C_{\mathrm{eq}}}{q_{\mathrm{eq}}}=\frac{C_{\mathrm{eq}}}{q_{\max}}+\frac{1}{K_{L}\cdot q_{\max}}$	q_eq_—amount adsorbed by zeolite at equilibrium, mg g^−1^ C_eq_—the equilibrium concentration in solution, mg L^−1^ q_max_—monolayer capacity of the adsorbent, mg g^−1^ K_L_—the Langmuir adsorption constant, L mg^−1^
R_L_	$R_{L}=\frac{1}{1+C_{0}\cdot K_{L}}$	R_L_—dimensionless C_0_—initial concentration in solution, mg L^−1^ K_L_—the Langmuir adsorption constant, L mg^−1^
Freundlich	$\ln\left(q_{\mathrm{eq}}\right)=\ln\left(K_{F}\right)+\frac{1}{n}\ln\left(C_{\mathrm{eq}}\right)$	q_eq_—amount adsorbed at equilibrium, mg g^−1^ C_eq_—the equilibrium concentration in solution, mg L^−1^ K_F_—the Freundlich constant, mg^1−n^ 1/n—heterogeneity factor, Ln g^−1^

**Table 4 molecules-28-00798-t004:** Freundlich and Langmuir parameters for the adsorption isotherms.

Adsorption Model						
Sample	Langmuir			Freundlich		
	q_max_ mg g^−1^	K_L_	R^2^	q_max_ mg g^−1^	K_F_	R^2^
NaP1_C	5.66	2.19	0.31	7.14	1.91	0.74
NaP1_FA	1.19	1.03	0.29	5.72	1.66	0.91

R^2^—coefficient value, q_max_—maximum adsorption capacity, K_L_—Langmuir model constant, K_F_—Freundlich model constant.

**Table 5 molecules-28-00798-t005:** Comparison of the adsorption capacities of different adsorbents in reaction with erythromycin.

Adsorbent	EA Concentration mg L^−1^	Adsorption Capacity mg g^−1^	Dosage g L^−1^	Reference
NaP1_C	1.0	17.75	2.0	This work
NaP1_FA	1.0	11.75	2.0	This work
MAC	65.0	248.91	1.55	[73]
MACC	60.0	178.52	0.01	[67]
BSA/Fe_3_O_4_		144.51		[69]
MWCNT	100	104.8	0.8	[70]
MoS_2_	4.0	57.10	1.0	[74]
WS_2_	4.0	190.31	1.0	[74]
PMG	200.0	286.0	0.35	[71]
ERY@MIP	8000	44.03		[72]
ZCA	20	17.76		[68]

MAC—magnetic activated carbon; MACC—magnetic activated carbon—chitosan; BSA/Fe_3_O_4_—magnetic composite microsphere; MWCNT—multi-walled carbon nanotubes; MoS_2_—molybdenum disulfide; WS_2_—tungsten disulfide; PMG—porous magnetic graphene; ERY@MIP—erythromycin molecularly imprinted polymer; ZCA—zeolite/cellulose acetate blend fiber.

**Table 6 molecules-28-00798-t006:** Values of the correlation coefficient (R^2^) for the kinetic models used.

Kinetic Model	Adsorbent	
	NaP1_C	NaP1_FA
pseudo-zero-order model	0.0008	0.1023
pseudo-first-order model	0.0082	0.0859
pseudo-second-order model	0.0073	0.062
BMG model	1	1

**Table 7 molecules-28-00798-t007:** Characteristics of wastewater used in adsorption experiments.

Parameter or Component	Method of Determination	Unit	Result ± Uncertainty
pH	potentiometric	-	7.6 ± 0.1
Electrical conductivity	conductometric	μS cm^−1^	1730 ± 87
Salinity	(at a reference temperature of 20 °C)	mg NaCl L^−1^	870 ± 44
Turbidity	conductometric	FAU	99 ± 10
Color	(at a reference temperature of 20 °C)	mg Pt L^−1^	77 ± 15
COD(Cr)	nephelometric	mg O_2_ L^−1^	390 ± 59
TOC	spectrophotometric	mg L^−1^	183 ± 27
Ammoniacal nitrogen	spectrophotometric	mg L^−1^	83 ± 8
Total phosphorus	spectrophotometric	mg L^−1^	10 ± 1
Phosphate	spectrophotometric	mg L^−1^	7.0 ± 0.7
Chlorides	spectrophotometric	mg L^−1^	160 ± 16
Sulphates (VI)	spectrophotometric	mg L^−1^	125 ± 13

## Data Availability

Not applicable.

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
