# Peer review of "Erythromycin Scavenging from Aqueous Solutions by Zeolitic Materials Derived from Fly Ash"

_molecules, 2023, doi:10.3390/molecules28020798_

Round 1
Reviewer 1 Report
The present paper deals the use zeolite (NaP1_FA) and zeolite-carbon composite 122 (NaP1_C) as potential adsorbents for the EA removal. The writing of the manuscript is poor and discussion is not appropriate. Thus, I am recommending it for rejection.
Q. Do not use any short notaion in abstract. For example,” Fly-ash based synthetic zeolite (NaP1_FA) and carbon-zeolite composite (NaP1_C) were 17 used to remove EA from aqueous solutions and real wastewater” Whatis EA? Did author define before?
Q.2. Abstract needs to be revised with a better presentation. The present form of the abstract cannot be considered as its lacks many details.
Q. 3. Author should aslo add one graphical abstract for better understanding of the work. Also, the title of the manuscript seems little with complex. Author must revise it.
Q. 4. “The properties of EA highly depend on the pH.” What kind of properties author looking that highly subjected to pH?
Q. 5. “Oxidation, electro-precipitation, mem-90 brane separation, coagulation-flocculation, evaporation, floatation, ion exchange have 91 been widely used, but these methods are insufficient to eliminate antibiotics from 92 wastewater [23,24,35] [23,24,35].” The reference cited twice, that need to be fixed.
Q. 6. The research gaps, novelty and objective of the study is missing. Why this study become essential? How this manuscript is different than other published one? “The novelty of this work involves the use of two different fly-ash based adsorbents 130 that show a very high efficiency of EA removal both for artificial and real wastewaters. 131 Moreover,” by stating simple line not make it different.
Q. 7. In the present study author used so many short notations but did not explain the full name. It would be better if author should add a separate page for all the used notation.
Q. 8. “Zeolite NaP1 was produced based on hydrothermal synthesis (for 24 hours 139 at 353 K) of fly ash with sodium hydroxide at atmospheric pressure.” Hour should be h. also, “Examination of samples’ surface area 182 and porosity was carried out on the basis of low temperature N2 adsorption/desorption 183 measurements (77 K) using Micromeritics ASAP 2020 instrument” N2 should be N2.
Q. 9. “ UPLC–MS/MS determination method” what is UPLC?
Q. 10. The discussin part in the results section is missing thoroughly. For example, “section 3.1.1” is just stating the obtained results. Author did not even try to explain why how etc.
Q. 11. All the figures need complete revision; the present form of the figure cannot be accepted.
Q. 12. Section 3.1.2 is just a report. Author did not try to explain the results and compared with similar finding’s.
Q. 13. Fig. 2 need complete revision with better representation. The surface area of adsorbent is 64 and removal efficiency is 94%. I did not convince. How?
Q, 14. Section 3.2.4. The effect of pH and dosage need better explanation with some comparative results.
Q. 15. Author must reduce the conclusion length.
Author Response
Q. 1. Do not use any short notaion in abstract. For example,” Fly-ash based synthetic zeolite (NaP1_FA) and carbon-zeolite composite (NaP1_C) were 17 used to remove EA from aqueous solutions and real wastewater” Whatis EA? Did author define before?
The authors thank the reviewer for the guidance, we have made changes to the text so that the EA abbreviation is already explained in the abstract. EA is: Erythromycin (line 17).
Q. 2. Abstract needs to be revised with a better presentation. The present form of the abstract cannot be considered as its lacks many details.
The authors thank the reviewer for the remark. We have included a new version of the abstract in the manuscript, which is as follows .
“Erythromycin (EA) is one of the antibiotics for which concentrations above standard levels in water and wastewater have been reported. Developing effective methods to remove EA from the aquatic environment is necessary, as the methods used so far are not very effective. In our study, we use fly ash (FA)-based zeolite materials, which have not yet been investigated as EA sorbents. We show the possibility of managing waste FA and, at the same time, using its transformation products for EA sorption. We investigate the efficiency of EA removal from experimental solutions and real wastewater. We report on the sorbents' mineral composition, chemical composition and physicochemical properties and the effects of adsorbent mass, contact time, initial EA concentration and pH on the EA removal. EA is removed within the first two minutes of the reaction with an efficiency of 99% from experimental solutions and 94% from real solutions. The maximum adsorption capacities were 314.7 mg g-1 for the fly-ash-based synthetic zeolite (NaP1_FA) and 363.0 mg g-1 for the carbon-zeolite composite (NaP1_C). A five-fold regeneration of the NaP1_FA and NaP1_C indicated no significant loss of adsorption efficiency. Our results indicate that the zeolitic materials effectively remove EA and can be investigated for removing other pharmaceuticals from water and wastewater.”
Q.3. Author should aslo add one graphical abstract for better understanding of the work. Also, the title of the manuscript seems little with complex. Author must revise it.
The authors thank the reviewer for the suggestion. We have added a graphical abstract. We have changed the title to: "Erythromycin scavenging from aqueous solutions by zeolites from fly ash." The title of the article is simple, unambiguous and focuses on the problem. Additionally we have added a keyword: carbon-zeolite composites.
Q.4. “The properties of EA highly depend on the pH.” What kind of properties author looking that highly subjected to pH?
We are sorry but your questions in the current form is not clear to us. Please specify exactly what do you want us to add here. The properties of EA may include different aspects e.g. formal charges, stability and conformation which highly depend on the pH. Some of these aspects have been raised in discussion on removal mechanisms which has been added as a new figure number 8.
Thank you for your remark. The pH of EA commercially available as an active drug substance (saturated solution) ranges from 8 to 10.5; and below pH 4 EA is destroyed and unstable [Lewis, RJ Sr.; Hawley's Condensed Chemical Dictionary, 15th Edition. John Wiley & Sons, Inc. New York, NY 2007, p. 510]. But this concerns finished medicinal products, and our goal was to check whether acidification or alkalization of the sample has a positive effect on the sorption of the antibiotic in the materials used. The manipulation of pH values was supposed to give an insight into the strength of the analyte-sorbent interaction. The article at the link https://www.ncbi.nlm.nih.gov/pmc/articles/PMC5485444/ presents a modified, more acid-stable form of EA to minimize the toxicity of the compound and increase its hydrophobicity, which improves the overall bioavailability of EA at the site of infection.
It follows that the results of our experiments taking into account a wide pH range may be beneficial from the point of view of water treatment and reduction of the toxic effects of EA residues in the aquatic environment.
Q. 5. “Oxidation, electro-precipitation, membrane separation, coagulation-flocculation, evaporation, floatation, ion exchange have been widely used, but these methods are insufficient to eliminate antibiotics from wastewater [23,24,35] [23,24,35].” The reference cited twice, that need to be fixed.
The authors thank the reviewer for the remark, we have made a correction in the text (line 96).
Q. 6. The research gaps, novelty and objective of the study is missing. Why this study become essential? How this manuscript is different than other published one? “The novelty of this work involves the use of two different fly-ash based adsorbents that show a very high efficiency of EA removal both for artificial and real wastewaters. Moreover,” by stating simple line not make it different.
The authors thank the reviewer for the suggestion. The following information has been added to the abstract:
„In our study, we use fly ash (FA)-based zeolite materials, which have not yet been investigated as EA sorbents. We show the possibility of managing waste FA and, at the same time, using its transformation products for EA sorption. We investigate the efficiency of EA removal from experimental solutions and real Wastewater”.
Unfortunately, the original text of the article did not include the information that such studies were conducted for the first time and that NaP1_FA and NaP1_C sorbents have never been used for erythromycin removal in the past. Therefore, according to a valuable suggestion, the following information was added to the text of the article: "The results presented in this paper on erythromycin removal were obtained for the first time using sorbents: NaP1_FA and NaP1_C."
Q. 7. In the present study author used so many short notations but did not explain the full name. It would be better if author should add a separate page for all the used notation.
The authors thank the reviewer for the suggestion, but in accordance with the Instructions to Authors we have explained the abbreviations directly in the text. „Acronyms/Abbreviations/Initialisms should be defined the first time they appear in each of three sections: the abstract; the main text; the first figure or table. When defined for the first time, the acronym/abbreviation/initialism should be added in parentheses after the written-out form.”
Q. 8. “Zeolite NaP1 was produced based on hydrothermal synthesis (for 24 hours at 353 K) of fly ash with sodium hydroxide at atmospheric pressure.” Hour should be h. also, “Examination of samples’ surface area and porosity was carried out on the basis of low temperature N2 adsorption/desorption measurements (77 K) using Micromeritics ASAP 2020 instrument” N2 should be N2.
The authors thank the reviewer for the remark. We have made a correction in the text (lines 130, 144 and 208).
Q. 9. “ UPLC–MS/MS determination method” what is UPLC?
The authors thank the reviewer for the remark. We have included the abbreviation of UPLC as Ultra-Performance Liquid Chromatography. UPLC or UHPLC are identical liquid chromatography techniques used to separate the various components present in mixtures. In 2004, Waters introduced a technique known as Ultra Performance LC (UPLC) using columns with less than 2 µm packing. As other vendors entered the market with similar instruments, UHPLC was coined as a way to refer to UPLC-like instruments. These two terms are understood as synonyms and are commonly used to describe the methodology we use.
Q. 10. The discussin part in the results section is missing thoroughly. For example, “section 3.1.1” is just stating the obtained results. Author did not even try to explain why how etc.
The authors thank the reviewer for the suggestion. Several new findings have been included and discussed in the manuscript. We have also added citations regarding other spectroscopic results in the section you have mentioned.
Q. 11. All the figures need complete revision; the present form of the figure cannot be accepted.
The authors thank the reviewer for the comment, however specific comments regarding figures have not been provided in your review. In our opinion the figures are clear, concise and show the most important data. Some minor changes have been made to the figures after remarks from other reviewers. Moreover, the figures were prepared according to guide for authors in terms of technical requirements. Please specify your remarks in your further review if necessary.
Q. 12. Section 3.1.2 is just a report. Author did not try to explain the results and compared with similar finding’s
The authors thank the reviewer for the suggestion. We have included a discussion and citations in section 3.1.2.
Wokusz, A., Zofka, A., Bandura, L., Franus, W., Effect of zeolite properties on asphalt foaming. Construction and Building Materials. 2017, 139
Zhou, Q., Xuguang, J., Qui, Q., Zhao, Y., Long, L., Synthesis of high-quality Na-P1 zeolite from municipal solid waste incineration fly ash by microwave-assisted hydrothermal method and its adsorption capacity. Science of the Total Environment. 2023, 855
Q. 13. Fig. 2 need complete revision with better representation. The surface area of adsorbent is 64 and removal efficiency is 94%. I did not convince. How?
The authors thank the reviewer for the question. The surface area is not always the crucial factor which leads to high efficiency of adsorbents. The crucial factors here include surface active sites which are a driving force for efficient adsorption. The discussion on mechanism has been provided in the manuscript and given in the newly prepare figure.
Q, 14. Section 3.2.4. The effect of pH and dosage need better explanation with some comparative results.
The authors thank the reviewer for the remark. The title of chapter 3.2.4 has been corrected to Regeneration studies.
Q. 15. Author must reduce the conclusion length.
The authors thank the reviewer for the suggestion. According to the suggestion the conclusion part was shortened. Only the most important findings have been highlighted and provided.
Reviewer 2 Report
The removal of Erythromycin from water is an important topic since the pollution of antibiotics are harmful to environment and human health. This manuscript reported the adsorptive removal of Erythromycin from water using fly-ash based synthetic zeolites and carbon-3 zeolite composites. The batch adsorption experiments were designed well, and the results were convicing.
1. In Figure 4. the adsorption process seems not reach equilibrium, since the maximum was still rising.
2. The maximum adsorption capacities (Table 5) were found to be 5.66 and 1.19 mg/g based on the Langmuir model, These values were very low and much lower than many previous adsorbents in literature. Such low adsorption capacity made the present materials not suitable for real wastewater treatments.
3. Why μg/g was used in the Fig. 5b.
4. As indicated in Fig.6, the present material exhibited much lower adsorption capacities than previous materials. Please discuss and explain this aspect.
5. Generally, the temperature, coexisting ions and DOM can affect the adsorption of antibiotics, please add these experiments.
6. Overall, this manuscript needs major revision before further consideration.
Author Response
1. In Figure 4. the adsorption process seems not reach equilibrium, since the maximum was still rising.
We agree with the reviewer's comment. However, our initial concentration range was already very broad (0.1 - 200 mg EA/L). Especially the maximum concentration (200 mg/L) is much higher from the EA concentrations encountered in real conditions. It should therefore be considered positive that, despite such a high initial EA concentration, the maximum saturation of the sorbents was not reached. To achieve a state of sorption equilibrium, even higher initial concentrations would have to be used, which would not make any environmental sense.
2. The maximum adsorption capacities (Table 5) were found to be 5.66 and 1.19 mg/g based on the Langmuir model, These values were very low and much lower than many previous adsorbents in literature. Such low adsorption capacity made the present materials not suitable for real wastewater treatments.
The reviewer recalls the results in Table 5 obtained from the Langmuir model. However, the current results of the maximum sorption volumes are higher. From the experimental data, the sorption capacity of EA on the NaP1_C and NaP1_FA is 7.14 and 5.72 mg/g, respectively. However, the main advantage of the sorbents we use is that they are fully regenerable, allowing the same portion of sorbent to be used repeatedly. In addition, the efficiency of EA sorption from solutions containing EA in quantities very close to these found in real environmental system is close to 100%. Therefore, we believe that the sorbents we have described can remove EA from real wastewater which has been confirmed in our studies.
3. Why μg/g was used in the Fig. 5b.
The authors thank the reviewer for the remark. This was corrected to μg/L. We have used values in μg/L to clearly show what was the drop in EA concentration depending on the adsorbent dose (suspension density). However, we have decided to include a right axis showing the adsorption efficiency in %. We hope this will help the reader to follow the results.
Fig 5b has been corrected. Thank you very much for your suggestion.
4. As indicated in Fig.6, the present material exhibited much lower adsorption capacities than previous materials. Please discuss and explain this aspect.
We are sorry but we cannot give an answer to the reviewer’s comment due to lack of specific remark. Please provide a more detailed comment – what materials do You specifically mean?
5. Generally, the temperature, coexisting ions and DOM can affect the adsorption of antibiotics, please add these experiments.
The authors thank the reviewer for the comment. The reviewer is right, the adsorption of antibiotics (or generally organic compounds) may be affected by temperature and the chemical composition of the aqueous environment. In our studies we have focused on basic experimental parameters including pH, dosage and concentration. However please note that we have included results of the experiment carried out for real wastewater. The wastewater characteristics are shown in Table 2 and clearly show its complex chemical nature including several inorganic components as well as a high TOC content. The tested zeolitic materials show high removal efficiency of EA present in such wastewater. This nicely shows the applicability of the studied materials for real aqueous system. Therefore, we have decided not to include more experimental results.
6. Overall, this manuscript needs major revision before further consideration.
The authors thank the reviewer for the comment. We have improved the manuscript by analyzing all remarks and remarks of other reviewers. In our opinion the manuscript has been significantly improved and the scientific content will provide valuable information for the readers.
The most important modifications we have included are:
1. We corrected the title and abstract of the article.
2. In section 2.1, we described the major difference between NaP1_FA and NaP1_C
3. In section 2.2. we described the method of determining surface charge density as a function of solution pH and point of zero chargé
4. In section 2.4.2, we added Figure 1. Research methodology flow chart
5. In section 3.1.1, we added a discussion of XRD and FTIR analysis
6. In section 3.1.2. we added a discussion of DTA/TG and carbon, hydrogen and, nitrogen elemental analysis
7. In section 3.2.2. we added a discussion of the effect of dosage, changed Figure 6b, and added information to Table 6
8. In section 3.2.2, we added Figure 7. Surface charge density of NaP1_C and NaP1_FA as a function of solution pH, and interpretation of the results.
9. In section 3.2.2. we added Figure 8. Possible interactions of erythromycin with zeolitic materials and its description.
10. We have adjusted the conclusions.
Reviewer 3 Report
Manuscript ID: molecules-2114418
Title: Erythromycin scavenging from artificial aqueous solutions and real wastewater by fly-ash based synthetic zeolites and carbon-zeolite composites
Comments: This work requires a major correction since the work done is fine but needs to express the scientific outcome of the proposed work. Authors are required to address these comments for the improvement of the paper.
· Authors are required to provide novelty in their research. Has the same research been carried out in the past?
· There is no need to include Table 1; it is preferable to leave it out of the revised manuscript.
· Include a flow chart of the methodology.
· It is preferable to incorporate a potential pictorial mechanism into the proposed model.
· The comparison table number 6 needs to incorporate some of the most recently reported works and should also include further information on how the suggested material relates to other materials.
· Figures 2, 5, and 6 need to be updated to ones that are more legible.
· “Removal of EA from wastewater…” Lines 424 through 426 must be justified.
· Conclusions should be more precise and short rather than lengthy discussion.
· Overall, check the language of the paper and verify the results.
Author Response
1. Authors are required to provide novelty in their research. Has the same research been carried out in the past?
The authors thank the reviewer for the suggestion. The following information has been added to the abstract:
„In our study, we use fly ash (FA)-based zeolite materials, which have not yet been investigated as EA sorbents. We show the possibility of managing waste FA and, at the same time, using its transformation products for EA sorption. We investigate the efficiency of EA removal from experimental solutions and real Wastewater”.
In addition, as suggested, the following information about the novelty of the research presented was included in the introduction:
„The novelty of this work involves the use of two different fly-ash based adsorbents that show a very high efficiency of EA removal both for artificial and real wastewaters. Moreover, we have shown that these materials can be easily regenerated and reused without significant decrease in adsorption efficiency. In addition, the con-ducted experimental study will undoubtedly increase the knowledge required for conducting further studies on a larger scale.”
Unfortunately, the original text of the article did not include the information that such studies were conducted for the first time and that NaP1_FA and NaP1_C sorbents had never been used for erythromycin removal in the past. Therefore, according to a valuable suggestion, the following information was added to the text of the article: "The results presented in this paper on erythromycin removal were obtained for the first time using sorbents: NaP1_FA and NaP1_C."
2. There is no need to include Table 1; it is preferable to leave it out of the revised manuscript.
The authors thank the reviewer for the suggestion. In our opinion the table showing chemical structure of EA is crucial for the reader to follow and understand the discussion part of the manuscript. Therefore, the authors decided to leave it as it is.
3. Include a flow chart of the methodology.
The authors thank the reviewer for the suggestion . Fig. 1. Research methodology flow chart has been added in section 2.4.2.
4. It is preferable to incorporate a potential pictorial mechanism into the proposed model.
The authors agree with the reviewer's suggestion. A model showing possible interactions of erythromycin with zeolitic materials has been prepared and included (Figure 8). It briefly shows two types of mechanism: (I) electrostatic interaction of EA with negatively charged surface of zeolites and (II) hydrophobic interaction of EA with carbon component present in the NaP1_C material. In particular, the latter mechanism is responsible for highly efficient adsorption.
5. The comparison table number 6 needs to incorporate some of the most recently reported works and should also include further information on how the suggested material relates to other materials.
The authors thank the reviewer for the suggestion. As suggested in Table 6, other materials have been added and, in addition, the results in Table 6 have been appropriately commented.
6. Figures 2, 5, and 6 need to be updated to ones that are more legible.
Figures 2, 5 and 6 has been updated to be more legible and clear.
The numbering of the figures has been updated:
Fig. 2 is now Fig. 3
Fig. 5 is now Fig. 6
Fig. 6 is now Fig. 9
7. “Removal of EA from wastewater…” Lines 424 through 426 must be justified.
The authors added a justification in the text of the article:
"The highest efficiency of EA removal from wastewater was obtained using zeolites with the smallest fraction of 200 μm (WWTP Stupava: EA 94.7 %, WWTP Devínska Nová Ves: EA 98.5 %). Such results confirm that zeolites can be used as sorbents to remove EA from wastewater."
8. Conclusions should be more precise and short rather than lengthy discussion.
The authors thank the reviewer for the suggestion , the conclusions part has been shortened.
9. Overall, check the language of the paper and verify the results.
The authors thank the reviewer for the comment. The manuscript has been additionally checked in terms of language by external editing service (English native experts).
The authors have checked all results and revised the drawings. In addition, the authors have added the pH - ZPC results in section 3.2.2.
Reviewer 4 Report
The manuscript deals with the characterization of the removal of EA from real wastewater using synthetic zeolites and carbon-zeolite composite. The adsorption process was studied at different pH and adsorbent concentrations, and was fit with several adsorption isotherms including Langmuir. The kinetics of the adsorption process was also studied. The study is comprehensive and detailed, and manuscript is clear and well-written and effective removal was observed. Hence, the manuscript is suitable for publication once the following questions/issues are addressed.
General questions/comments:
- The colloidal properties (zeta potenial, size, aggregation) of the adsorbent themselves were not characterized. What is the aggregation status of the adsorbents at the pH's used since some zeolites have an isoelectric point at a certain pH? This generally changes the dominant interaction forces involved, which might explain the difference in the effect of pH between the two adsorbents.
- What is the difference between the two zeolites, if both are based on fly-ash, according to the synthesis procedure?
- In XRD and FTIR, literature comparison needed.
- In the line 161 and line 170, the concentration refers to EA, but is a bit ambiguous. For clarity, the text shall preferably be modified to clearly state it refers to EA.
- In Figure 5(b), the EA concentration on the y-axis is doubtedly in ug/L, not ug/g, as described in the text.
- Why wasn’t the effect of the pH probed in the neutral regime? And at what pH was the regeneration studied?
- Section 3.2.4 "The effect of pH and dosage" is not correctly titled, it discusses the regeneration.
- Why the initial cycle of NaP1_FA was low?
The manuscript have several style-related issues such as:
- Chemical structure in Table 1 (incomplete)
- Undefined terms such CHN and STA
- Line 84 (omit dash)
- Line 128 (DTA/TG not STA/TG)
- CO2, R2, MoS2, WS2, SiO2, Al2O3, and N2 (superscript or subscript)
- Line 189: The abbreviation of BJH is first mentioned without definition?
- Lines 185/239: 105°C and 35°2θ (no spaces)
Author Response
1. The colloidal properties (zeta potenial, size, aggregation) of the adsorbent themselves were not characterized. What is the aggregation status of the adsorbents at the pH's used since some zeolites have an isoelectric point at a certain pH? This generally changes the dominant interaction forces involved, which might explain the difference in the effect of pH between the two adsorbents.
The authors thank the reviewer for the valuable remark. Additional measurements have been performed. The authors have added a methodology of the pH - ZPC determination in section 2.2 and the results and discussion of pH - ZPC in section 3.2.2.
The dependency of NaP1_FA and NaP1_C surface charge density versus solution pH is presented in Figure 7. Potentiometric titration indicated that the pH - ZPC of both sorbents is about 3.0. This result means that at pH 3.0, their surface is characterized by zero surface charge. At pH < 3.0, the solid surface is positively charged. In turn, at pH > 3.0 - it is negatively charged. Above pH 3.0, the surface negative charge density of NaP1_FA is almost twice higher as compared to the NaP1_C at the same pH values. This relationship is correlated with the sorption efficiency of EA, which is twice lower for the NaP1_FA in alkaline media. The EA has a pKa of 8.89, indicating its molecules are positively charged at pH < 8.89. This explains the relatively high sorption on negatively charged zeolite materials under these pH conditions. However the electrostatic attraction seems to play a secondary role in the removal process as the NaP1_C material shows higher efficiency. This clearly indicates the dominant role of hydrophobic interactions between carbon component present in the NaP1_C material and EA organic structure. The latter is rich in methyl and methylene groups which give the hydrophobic character. Such mechanism is independent of the solution pH. The more hydrophilic nature of the NaP1_FA than the NaP1_C results lower adsorption. On the one hand, the negative surface charge of NaP1_FA is twice that of NaP1_C, resulting in lower sorption. On the other hand, the carbon in the composite is responsible for hydrophobic interactions, resulting in less pH-dependent EA sorption on NaP1_C than on NaP1_FA.
2. What is the difference between the two zeolites, if both are based on fly-ash, according to the synthesis procedure?
The authors thank the reviewer for the question. The main difference is the carbon content between the two zeolitic materials. The Na-P1_C (zeolite-carbon composite) material was hydrothermally synthesized by using carbon-rich fly-ash. In turn the Na-P1_FA material was obtained by hydrothermal synthesis from a typical fly ash with low carbon content. The carbon content of the two materials after synthesis is different, with 5.53% wt. for the NaP1_FA and 42.19% wt. for the NaP1_C. Due to high content of carbon in the Na-P1_C material its surface chemistry was much different. In majority of our experiments this material showed higher compatibility with hydrophobic organic compounds. As a result its adsorption capacity was higher than that observed for the Na-P1_FA.
Text included in the manuscript: The main difference between NaP1_FA and NaP1_C is the use of fly ash with a different unburned carbon content for their synthesis. To synthesise NaP1_FA, F-class fly ash from coal combustion at the Rybnik power plant in Poland, which contains < 5% unburned carbon, was used [54]. For the synthesis of NaP1_C, high carbon fly ash (HCFA) collected from Janikowo Thermal Power Station (Janikowo, Poland) was used [57], which contains about 40% unburned carbon. The carbon content of the two materials after synthesis is also different, with NaP1_FA at 5.53% and NaP1_C at 42.19%.
3. In XRD and FTIR, literature comparison needed.
Comparison to the literature has been added in section 3.1.1.
4. In the line 161 and line 170, the concentration refers to EA, but is a bit ambiguous. For clarity, the text shall preferably be modified to clearly state it refers to EA.
The authors thank the reviewer for the suggestion, we have made changes to the text so that the concentration refers to EA.
5. In Figure 5(b), the EA concentration on the y-axis is doubtedly in ug/L, not ug/g, as described in the text.
The authors thank the reviewer for the remark. This was corrected to μg/L. We have used values in μg/L to clearly show what was the drop in EA concentration depending on the adsorbent dose (suspension density). However we have decided to include a right axis showing the adsorption efficiency in %. We hope this will help the reader to follow the results.
6. Why wasn’t the effect of the pH probed in the neutral regime? And at what pH was the regeneration studied?
The authors thank the reviewer for the question. The NaP1_FA and NaP1_C materials alkalize the whole system. They have an alkaline pH and, when added to aqueous EA solutions, the pH increases to a value of 8-9. The authors did not keep the pH at 7.0. The pH of the solutions after desorption ranged from 6-7. Ethanol was used for desorption.
7. Section 3.2.4 "The effect of pH and dosage" is not correctly titled, it discusses the regeneration.
The authors thank the reviewer for the suggestion. The title of section 3.2.4. has been corrected to Regeneration studies.
8. Why the initial cycle of NaP1_FA was low?
The authors thank the reviewer for the question. The adsorption in the first cycle is indeed lower, however please note larger error bars which indicate a higher error attributed to HPLC analysis of EA. Nevertheless the further cycles of adsorption/desorption clearly confirm very high adsorption after subsequent regeneration steps. This shows applicability of the material for the removal of EA in real environmental systems.
The manuscript have several style-related issues such as:
9. Chemical structure in Table 1 (incomplete)
The authors thank the reviewer for the comment. We have inserted the complete EA structure in Table 1 .
10. Undefined terms such CHN and STA
The authors thank the reviewer for the remark. We have defined the abbreviations.
CHN - carbon, hydrogen and nitrogen elemental analysis; clarification has been included in the text, section no. 2.3.
STA - Simultaneous Thermal Analysis, clarification has been added in the text, section no. 3.1.2.
11. Line 84 (omit dash)
This has been corrected appropriately.
12. Line 128 (DTA/TG not STA/TG)
The authors thank the reviewer for the correction. We have changed it to DTA/TG.
13. CO2, R2, MoS2, WS2, SiO2, Al2O3, and N2 (superscript or subscript)
The authors thank the reviewer for checking our work in detail we have corrected it throughout the text:
CO2, R2, MoS2, WS2, SiO2, Al2O3, and N2
14. Line 189: The abbreviation of BJH is first mentioned without definition?
The authors thank the reviewer for bringing this to our attention. The authors have corrected the mistake. The manuscript text explains the BJH meaning when it was first used.
Lines 188 – 191: The mesopores volume (Vmeso) was determined with the use of BJH method. The pore-size distributions were obtained from the desorption branch of the isotherm using the Barrett–Joyner–Halenda (BJH) procedure.
Has been changed to: The mesopores volume (Vmeso) was determined with the use of the Barrett–Joyner–Halenda (BJH) method. The pore-size distributions were obtained from the desorption branch of the isotherm using the BJH procedure.
15. Lines 185/239: 105°C and 35°2θ (no spaces)
The authors thank the reviewer for the correction. We have added a missing space.
Round 2
Reviewer 1 Report
Authour has now revised the maniscript nd improved the quality of manuscript. thus it can be accepte.
Reviewer 2 Report
Accept
Reviewer 3 Report
The revised manuscript can be accepted.